# SANE: Specialization-aware neural network ensemble

## Abstract

Real-world data is often generated by some complex distribution, which can be approximated by a composition of multiple simpler distributions. Thus, it is intuitive to divide complex model learning into training several simpler models, each of which specializes in one simple distribution, called model specialization in this paper. Ensemble learning is one way to realize specialization, and has been widely used in practical machine learning scenarios. Many ensemble methods propose to increase diversity of base models, which could potentially result in model specialization. However, our studies show that without explicitly enforcing specialization, pursuing diversity may not be enough to achieve satisfactory ensemble performance. In this paper, we propose SANE — an end-to-end ensemble learning method that actively enforces model specialization, where base models are trained to specialize in sub-regions of a latent space representing the simple distribution composition, and aggregated based on their specialties. Experiments in several prediction tasks on both image datasets and tabular datasets demonstrate the superior performance of our proposed method over state-of-the-art ensemble methods.

## 1 Introduction

Real-world data distribution could be complex in most cases and people usually approximate it by a composition of several simpler distributions (Xie et al., 2016; Yang et al., 2016; Tsai et al., 2020). Intuitively, to fit the complex data distribution conveniently, we can divide the complex model learning process into training several simpler models, each of which specializes in one simple distribution. In this paper, we call this idea model specialization.

Ensemble learning is one of the most effective methods to leverage model specialization. Originated from decades ago (Hansen & Salamon, 1990), ensemble method (Schapire, 1990; Breiman, 1996; Zhou et al., 2002) has been proven effective for practical machine learning tasks in various scenarios including computer vision (Huang et al., 2017), natural language processing (Shazeer et al., 2017) and tabular data mining (Liu et al., 2020).

Many works conduct ensemble learning through improving diversity across base models, which has been shown to reduce the variance of the combined predictions and improve generalization (Lee et al., 2015; Zhou et al., 2018). The recent methodologies pursuing model diversity mainly through incorporating randomization in data sampling (Breiman, 1996; Ho, 1995) or model training (Srivastava et al., 2014; Lakshminarayanan et al., 2016). Some other works propose sequential optimization such as boosting (Freund, 1995; Chen & Guestrin, 2016; Ke et al., 2017) and snapshot ensemble (Huang et al., 2017). Other methods, e.g., model decorrelation (Zhou et al., 2018) and diversity enhancement (Zhang et al., 2020), explicitly encourage model diversity through incorporating additional objectives. However, overvaluing diversity may hurt the ensemble effectiveness and the trade-off between the performance and diversity of base models still remains an open problem (Rame & Cord, 2021; Fort et al., 2019; Masegosa, 2019).

Moreover, without explicitly enforcing model specialization, pursuing model diversity does not provide guarantees of each model's specialization on different simple distribution in the composition. The last two columns in Figure 1 visualize the specialty of each model by plotting the data regions where the base model makes correct predictions. It shows that neither the ensemble of (b) randomly initialized base models nor (c) diversified base models produce models with different specialties.

Specifically, although the method in (c) enforces diversity and shows low correlation among base models' outputs, their data regions with correct predictions are very similar as shown in the last two columns in Figure 1, resulting in inferior performance after ensemble. Besides, simply putting specialized base models together does not necessarily bring promising ensemble performance. We need to know each base model's specialty over different samples and aggregate their outputs based on their specialty. As shown in Figure 1 (d), although the overall accuracy of the two base models are low, the ensemble model performs surprisingly well (100% accuracy in (d)) as long as 1) each base model specializes in unique sub-regions of the latent space and 2) the ensemble model knows which sub-regions each base model specializes in, demonstrating the effectiveness of specialization in ensemble learning. More detailed analyses of Figure 1 are presented in Section 2.

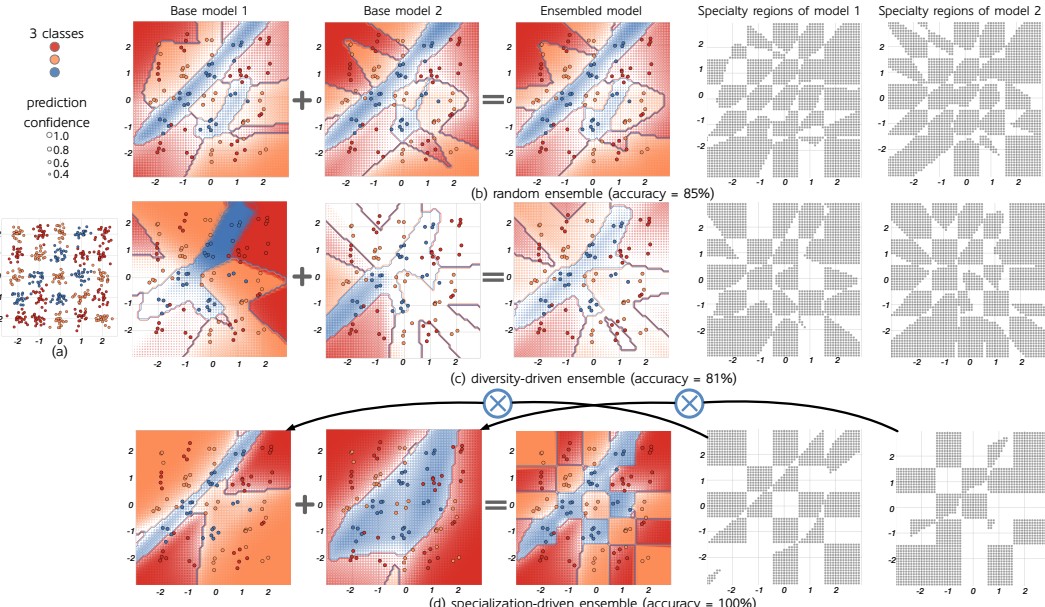

**Figure 1:** Comparison of three different ensemble methods on the synthetic checkboard dataset (a). The first three columns in (b) (c) (d) are decision boundaries of two base models and the final ensemble, resp., while the last two columns indicate the samples by gray dots for which each base model makes correct predictions. (b) is averaging ensemble of models with randomly initialization and trained independently. (c) introduces a negative correlation (diversity) loss into each base model's loss function on the basis of (b). (d) uses the models' confidence of true labels to weight the training loss and aggregate the model predictions accordingly.

In this paper, we propose an end-to-end ensemble learning method that actively enforces model specialization by training models to specialize in specific simple distributions, and aggregates them based on their specialties. Firstly, we represent the composition of simple distributions by mapping them into a unified latent space, where samples from each data distribution form a locally clustered sub-region. This paper proposes to enforce each base model to specialize in one sub-region, and we call this region the base model's corresponding *specialty region*. The base models' specialty regions are defined by a set of learnable anchor points, each of which corresponds to the center of the specialty region, called *specialty anchor*. During training, base models' parameters and their corresponding specialty anchors are learned simultaneously. Each base model's specialty is enforced by encouraging model to focus more on samples in the vicinity of its specialty anchor. Finally, during prediction, based on the correlation between models' specialty anchors and the samples in the latent space, our method is able to automatically estimate each base model's specialty over testing samples, and the outputs of the base models are adaptively aggregated based on their specialty over different testing samples. To demonstrate the effectiveness of the proposed method, we conduct extensive experiments in prediction tasks on two tabular datasets and two image classification datasets. The comprehensive empirical studies show that 1) coupling specialized base models and specialization-aware aggregation can benefit ensemble learning and 2) the proposed specialization-aware neural network ensemble method can achieve superior performance compared with the state-of-the-art ensemble methods in various prediction tasks.

## 2 PRELIMINARIES AND SYNTHETIC ANALYSIS

### 2.1 PRELIMINARIES

In general, ensemble learning consists of two tasks: training multiple base models and aggregating their predictions. We use $\{\mathcal{M}_k\}_{k=1}^K$ to denote a collection of $K$ base models. For a given instance $\mathbf{x}_n \in \mathbb{R}^d$ and its label $y_n \in [1, C]$, each base model learns a mapping $\mathcal{M}_k : \mathbb{R}^d \longmapsto \mathbb{R}^C$, which produces logits for $C$ classes. Base models' predictions are then aggregated in a weighted manner. The likelihood is written as $p(y_n \mid \mathbf{x}_n) = \sum_{k=1}^K w_k \, p(y_n \mid \mathbf{x}_n; \theta_k)$ s.t. $\sum_{k=1}^K w_k = 1$, where $\mathcal{M}_k$ is parameterized by $\theta_k$ and $w_k$ denotes the weight for aggregating $k$-th model prediction.

**Objectives of ensemble learning.** To maximize the likelihood, most previous studies attempt to promote $\sum_{k=1}^K p(y_n \mid \mathbf{x}_n; \theta_k)$ (true confidence of $\mathcal{M}_k$, for brevity), with fixed model weights on all samples. To this end, some works train models based on random initialization to reduce prediction variance and generalization errors, while others explicitly quantify diversity to reduce the correlation of errors between models. However, there is always a problem about the trade-off between diversity and accuracy in this scenario, since the weighted majority correctness among selected models determines the correctness of the ensemble. One way to avoid the conflict between adopting diverse models and boosting the overall performance is to use sample-specific model weights to aggregate models' predictions. Intuitively, if a base model performs much better than any other base models over a specific sub-region of the dataset containing one or more samples (i.e., a base model specializes in a sub-set of the dataset), it should be assigned with higher weight in this sub-region. As a result, we define the *specialized model* as follows.

**Definition 2.1.** *(Specialized model) We define the specialized model $\mathcal{M}_k$ of a sample $(\mathbf{x}_n, y_n)$ with two aspects: (1) $k = \arg\max_{k'} p(y_n \mid \mathbf{x}_n; \theta_{k'})$, and (2) $y_n = \arg\max_c p(c \mid \mathbf{x}_n; \theta_k)$ where $c \in [1, C]$ denotes the class label. If no model satisfies the above conditions, then $\mathbf{x}_n$ is not specialized by any model. That is, the specialized model is the one that works correctly and has the highest prediction confidence on a given sample among all the base models.*

### 2.2 SYNTHETIC ANALYSIS

In this section, we conduct the experiment with three ensemble solutions including a random initialization based method and a diversity-based ensemble method, and then we illustrate how specialization can benefit ensemble. A $5 \times 5$ checkerboard synthetic dataset containing 25 Gaussian blobs shown in Figure 1(a) is used for analysis.

Two widely used ensemble methods are selected for this experiment. Firstly, following the classical random ensemble, base models with different weight initialization are trained independently and aggregated averagely. On the basis of the random ensemble, we further implement a diversity-driven ensemble by incorporating a diversity loss (Li et al., 2012) into the error function of each base model to encourage base models to perform differently on the same sample. To illustrate how different base models and their ensembles make predictions, we draw their decision boundaries and graphs showing their individual specialty regions, which for a single model is where it makes the correct prediction. A multi-layer perceptron (MLP) with one hidden layer and 20 neurons serves as the base model of ensembles through this section. For simplicity of visualization and analysis, we maintain two base models for each ensemble method.

**Random ensemble v.s. diversity-driven ensemble.** The results of these two methods are, respectively, shown in Figure 1(b) and (c). From random ensemble results in Figure 1(b), the base models are *similar in performance and decision boundary* thus resulting in their limited ability for fitting the complex distribution with ensemble. One question may be raised: will enforcing the base learners to predict differently on the same samples by incorporating diversity loss be sufficient to solve the above issue? From Figure 1(c), base models' specialty regions for the diversity-driven ensemble are similar to that for the random ensemble, despite the fact that the prediction confidence of the base model diverges significantly across different sample regions. This suggests that diversity may not be sufficient to encourage the base model towards specialized on more samples, thus failing to improve ensemble performance. We can also observe that ensemble fails on some samples correctly predicted by a base model, for such a model we define as an *unexpressed specialist*.

**Definition 2.2.** *(Unexpressed specialist) Given the **specialized** model $\mathcal{M}_k$ on a sample $(\mathbf{x}_n, y_n)$, we define $\mathcal{M}_k$ as an unexpressed specialist if $y_n \neq \arg\max_c \sum_{i=1}^{K} w_i \, p(c \,|\, \mathbf{x}_n; \theta_i)$ where $c \in [1, C]$.*

**Specialization-aware ensemble.** In terms of diversity-driven results, the pursuit of diversity does not lead to model specialization. To verify whether a specialization-driven approach is better than the diversity-driven approach, we further use true confidence of model prediction on a sample (i.e., $p(c = y_n | \mathbf{x}_n; \theta_k)$) to explicitly quantify its specialization and conduct specialization-aware ensemble learning. As for training, we train the two base models while weighting the classification losses of each base model with the normalized true confidence. As a result, for model inference and ensemble, averaging their predictions only yields an accuracy of 71.0%, whereas an ensemble weighted by the true confidence of the base models yields an accuracy of 100.0%. It clearly illustrates that, as long as the specialization has been cautiously tackled, specialization-aware ensemble over weak base models will significantly improve the ensemble performance. As shown in Figure 1(d), base models' specialty regions overlap less, with more samples having a specialized model compared with the first two methods. Both specialization-aware training and output aggregation based on model specialty could contribute to the result that improves the accuracy from 71.0% to 100.0%. We verify that both factors are equally important by conducting an ablation study on the diversity-driven ensemble, where we assign the ensemble weights in (c) by the true confidences of each base model as we do in (d). However, the test accuracy is not improved (still 81%) by such a design.

Therefore, ideally, our target is to have an expressed specialist for each sample by training specialized models and aggregating based on their specialty accordingly. To this end, a problem with the current specialization-driven ensemble is that the base models tend to have overconfidence in the samples they are not specialized in (outside their known distributions) (Hein et al., 2019). These overconfident yet incorrect predictions tend to induce the presence of unexpressed specialists. Thus, not only do we need the models to be specialized in different specialty regions, but we also want these specialists to have a global perception to avoid overconfidence.

As opposed to specialized, we also try to jointly train the sum of base models by directly minimizing the cross entropy between ground truth and the final ensemble result $\frac{1}{K} \sum_{k=1}^{K} \mathcal{M}_k(\cdot)$. Surprisingly, from our experiments in Section 4.3 and evidences in other study (Allen-Zhu & Li, 2020), the performance of the trained large model is even worse than that of the single model. It shows that in this way we can actually give more loss weights in back-propagation for models with higher true confidence, i.e., it leads to specialization of models. However, without specialization-aware aggregation, the averaged ensemble result turns out to be inferior.

## 3 SPECIALIZATION-AWARE NEURAL NETWORK ENSEMBLE

We propose Specialization-Aware Neural Network Ensemble (SANE), an adaptive approach that learns to enforce the base models to specialize in sub-regions of the latent space representing the complex data distribution. In contrast to previous work on trading-off diversity and performance, our goal is to help the base models develop their own specializations and then aggregate their predictions based on their specialty. In this section, we will first state the solution structure and then present the details of our design.

### 3.1 OVERALL FRAMEWORK

We aim to train an ensemble of base models $\{\mathcal{M}_k\}_{k=1}^{K}$, each of which is parameterized by $\theta_k$ and gives its own prediction for the labels $y_n$ given input $\mathbf{x}_n$. To aggregate the predictions of the base models, an ensemble network $\mathcal{E} : \mathbb{R}^d \longmapsto \mathbb{R}^K$ parameterized by $\psi$ predicts the weights of the $K$ base models given the current input. $\mathbf{A} = [\mathbf{a}_1, \ldots, \mathbf{a}_K]$ is a set of learnable latent parameters of the ensemble network, and $\sum_{k=1}^{K} p(k \,|\, \mathbf{x}_n; \mathbf{A}, \psi) = 1$. As a result, the likelihood is written as:

$$p\left(y_n \,|\, \mathbf{x}_n; \mathbf{A}, \psi, \{\theta_k\}_{k=1}^{K}\right) = \sum_{k=1}^{K} p\left(k \,|\, \mathbf{x}_n; \mathbf{A}, \psi\right) p\left(y_n \,|\, \mathbf{x}_n, k; \theta_k\right). \tag{1}$$

Here, $p(k \,|\, \mathbf{x}_n; \mathbf{A}, \psi)$ can be regarded as the predictive confidence of $\mathcal{M}_k$ being a proper model for $\mathbf{x}_n$. For model $\mathcal{M}_k$, its prediction confidence of grounded label $p(y_n \,|\, \mathbf{x}_n, k; \theta_k)$ can be seen as a measure of appropriateness.

### 3.2 DESIGN

The whole network consists of two parts: a pool of base models $\{\mathcal{M}_k\}_{k=1}^K$ and an ensemble network $\mathcal{E}$. In this paper, $\mathbf{A}$ in ensemble network is defined as a set of anchors in a latent space of the dataset distribution, which will guide the base models towards their own specialty regions in the latent space. During training, base models are enforced to be more specialized on the vicinity of their corresponding specialty anchors, and are encouraged to be better at what they might already be skilled in. Ultimately, the whole sample space can be covered by specialty regions of base models, whose specialties are well captured by the specialty anchors, enabling a specialization-aware aggregation. In this section, we detail in turn the design of training diversified base models, the evolution of model specialization, and specialization-aware model aggregation.

#### 3.2.1 TRAINING DIVERSIFIED BASE MODELS

For many tasks, it is difficult to train a model to be excellent over the whole sample space. Therefore, we would like to partition the entire sample space into multiple local regions and train each base model to be a specialist in one region. Thereafter they can be aggregated together to produce a feasible solution.

In ensemble learning, it is common to partition samples before feeding them into different base models, i.e., to partition them randomly (Breiman, 1996) or according to their original feature space (Zhou et al., 2018). Either of these two ways would lead to a reduction in the number of samples for training each base model, which affects the performance of the model (especially for deep neural networks). Moreover, since neural networks can be overconfident in samples outside their known distributions, reducing samples will have a further influence on the ensemble performance, as discussed in Section 2. Besides, it is possible that segmenting samples through the raw feature space will not well capture valid information between model specialization and sample characteristics.

In contrast, we propose to locate base models' specialty regions in the latent space rather than in the raw feature space, since the latent space can provide meaningful representations (Gelada et al., 2019). In addition, the specialty regions of the models should be explored and learned in an adaptive manner to avoid inappropriate locking (Zhou et al., 2018). Through exploration, these base models not only focus on local regions, but also gain global perception. For these reasons, we introduce specialty anchors to guide the traversal of the latent space.

**Specialty anchors** are learnable representatives of simpler distributions in the latent space, the composition of which can derive a complex distribution to characterize the complexity of real-world data. Then, we can learn the base models, each of which is specialized in a specialty region represented by the corresponding specialty anchor. However, one challenge arises when relying on the specialty anchors for ensemble learning: how to capture the relationship between each sample and the specialty anchors to obtain the ensemble weights? To address this issue, we propose an attention-based method to guide the ensemble network to adaptively weigh the base models. In addition, we expect the specialty regions of the base models to cover the entire sample space, which may place a high requirement on exploration guided by these specialty anchors. Therefore, we further resort to the help of explicit specialization measures, which will be detailed in Section 3.2.2.

Explicitly, the guidance is achieved by attentional weighting the model loss w.r.t. the specialty anchors on each sample. Given $\mathbf{x}_n$, an encoder $\mathcal{F}$ computes a nonlinear mapping of $\mathbf{x}_n$ as $e_n$ with the same embedding dimension as $\mathbf{A}$. To capture the relations between $\mathbf{x}_n$ and anchors $\{\mathbf{a}_k\}_{k=1}^K$, we compute the pairwise attention as $\{\text{Softmax}(\frac{\mathcal{Q}(\mathbf{a}_k)(\mathcal{K}(e_n))^\top}{\sqrt{d_h}})\mathcal{V}(e_n)\}_{k=1}^K$, where $\mathcal{Q}, \mathcal{K}, \mathcal{V}$ are three projection functions each of which is implemented as one layer MLP and $d_h$ is the attention head number. These attention results are further fed into the scale-transfer layer, and then these $K$ values are processed by softmax operation $\text{Softmax}(\mathbf{z})_k = e^{z_k}/\sum_{j=1}^K e^{z_j}$ to produce the weights $\mathbf{w} = [w_1, \ldots, w_K] \in \mathbb{R}^K$ corresponding to the $K$ models. That is, in contrast to hard assignment where one sample is associated with a single model or a subset of models, in our ensemble framework, one sample is associated with all models simultaneously. The loss of the base models can be written as

$$\mathcal{L}_b = \sum_{k=1}^K \left( w_k + \frac{\tau_b}{1 + \log(T)} \right) \cdot \mathcal{H}\left(\mathcal{M}_k\left(\mathbf{x}_n\right), \mathbf{y}_n\right), \tag{2}$$

where $\tau_b$ is the temperature parameter, $T$ is the number of iterations, and $\mathcal{H}$ is the cross-entropy loss.

### 3.2.2 THE EVOLUTION OF MODEL SPECIALIZATION

Given each base model, we propose a learning-based method to know its specialty region. As mentioned above, $w_k$ is the learned confidence for leveraging the $k$-th base model to make predictions for each sample. Also, during training, we can observe the true confidence of each base model for each sample, denoted as $o_k = p(y_n \,|\, \mathbf{x}_n, k; \theta_k)$. Thus, $w_k$ should be close to $o_k$. In other words, for a pair $(\mathbf{x}_n, y_n)$, the weight on the model with higher true confidence should be greater than the weight on the model with lower true confidence. To facilitate the training of the ensemble network, we introduce the confidence loss as follows:

$$\mathcal{L}_c = -\sum_{k=1}^{K} \left( o_k \cdot \ln(w_k) + (I - o_k) \cdot \ln(1 - w_k) \right). \tag{3}$$

This work is not the first to favor model specialization. Zhou et al. (2018) attempted to assign a subset of samples to each base model based on their current expertise. However, such design may limit the performance of the base models for two reasons. First, the specialization of each base model is limited to a set of assigned samples, which prevents each base model from exploring other specialty regions. Second, it will lead to overfitting and overconfident issues, as discussed in Section 2. By incorporating specialty anchors and $\mathcal{L}_c$, we expect the ensemble network to gradually master predicting whether a base model is potentially specialized in certain samples. Therefore, in addition to focusing on the current specialty regions of each base model, specialty anchors and $\mathcal{L}_c$ also help to explore other regions where each base model may potentially be specialized in.

### 3.2.3 SPECIALIZATION-AWARE MODEL ENSEMBLE

While the $\mathcal{L}_c$ only regularizes the normalized true likelihood of base models, ensemble can still fail if all base models' true confidence is low. Thus, we further include an ensemble loss to optimize the likelihood of final target as follows:

$$\mathcal{L}_e = \mathcal{H}\left( \sum_{k=1}^{K} \text{Softmax}(\mathbf{w} + \frac{\tau_e}{1 + \log(T)})_k \cdot \mathcal{M}_k(\mathbf{x}_n), y_n \right), \tag{4}$$

where $\tau_e$ is the temperate. It is worth noting that $\mathcal{L}_e$ does not update the parameters of $\mathcal{M}_k$, only the ensemble network. As mentioned in Section 2, ensemble loss gives more weight to models with higher confidence in back-propagation. This collides with the effect of $\mathcal{L}_c$, which is intended to slowly approach the true confidence. In Section 4.3, we will perform an ablation study on it.

All three losses $\mathcal{L}_b$, $\mathcal{L}_c$ and $\mathcal{L}_e$ are combined to accomplish our goal. For each loss, we assign an independent loss weights $s_b, s_c$ and $s_e$ as hyper-parameters, so that the total loss is given by

$$\mathcal{L} = s_b \mathcal{L}_b + s_c \mathcal{L}_c + s_e \mathcal{L}_e. \tag{5}$$

## 4 EXPERIMENTS

In this section, we seek to answer research questions as follows. (**RQ1**) Is SANE superior to state-of-the-art ensemble methods, especially compared with diversity-driven methods? (**RQ2**) Is it better to encourage models to actively specialize at specific regions than statically assigning sample subsets with diversity regularization? (**RQ3**) How does each proposed module in SANE affect the final performance? (**RQ4**) Can SANE encourage models to be specialized over different sample regions?

### 4.1 DATASETS AND BASELINES

We briefly describe the benchmark datasets, baselines and the implementation in this section. For all classification tasks we use **accuracy** to evaluate, and in addition Area under ROC Curve (**AUC**) has also been adopted as the evaluation metric for binary classification. A detailed description of the experimental setup can be found in Appendix A.1.

#### 4.1.1 TABULAR DATASETS

***Higgs Boson*** (Baldi et al., 2014) and ***Blastchar*** (Kaggle, 2019) are two binary classification benchmarks. Higgs Boson contains 10.5M training samples and 0.5M test samples. Blastchar has 7,043

samples, from which we randomly sample 20.0% of the dataset as the test data. We split 12.5% of the training samples from both datasets for validation.

**Baselines.** Gradient boosting decision trees (GBDT) are widely used in tabular data. Therefore, we compare two state-of-the-art algorithms, **XGBoost** (Chen & Guestrin, 2016) and **LightGBM** (Ke et al., 2017). We also conduct a comparison with the deep neural network based model **TabNet** (Arık & Pfister, 2020) which is proven to achieve state-of-the-art results on tabular data. Additionally, we benchmark with random ensemble, in which the base models are trained independently with random initialization and the outputs are averaged for inference.

**Table 1:** Evaluation of tabular datasets. § and ‡ indicate results taken from Anghel et al. (2019) and Arık & Pfister (2020), respectively. The taken results of XG-Boost and LightGBM for Higgs Boson are evaluated on the same test set as ours, with exhaustive hyper-parameter optimization.

| Tabular dataset | Higgs Boson | | Blastchar | |
|---|---|---|---|---|
| | Accuracy | AUC | Accuracy | AUC |
| XGBoost | - | 83.53%§ | 80.06% | 83.06% |
| LightGBM | - | 85.73%§ | 80.70% | 84.53% |
| TabNet | 78.84%‡ | - | 80.48% | 84.21% |
| Single model | 78.49% | 87.04% | 81.48% | 85.45% |
| Random ensemble | 78.84% | 87.36% | 81.41% | 85.57% |
| SANE | **79.22%** | **87.74%** | **81.83%** | **85.78%** |

**Table 2:** Classification top-1 accuracy of different ensemble methods on CIFAR-10 and CIFAR-100 with $K = 3$. The * indicates result is taken from the original paper, while the § represents from Zhang et al. (2020).

| Backbone | Method | CIFAR-10 | CIFAR-100 |
|---|---|---|---|
| | | Accuracy (top-1) | |
| VGGNet-19 | Single model | 93.21% | 73.13% |
| | Random ensemble | 93.80%§ | 73.81%§ |
| | MOE method | 92.87% | 70.46% |
| | Fast ensemble | 93.29% | 74.53% |
| | Snapshot ensemble | 94.53% | 74.60% |
| | MEAL | 94.45%* | - |
| | Diversified ensemble | 94.91%* | 74.94%* |
| | SANE | **95.04%** | **77.64%** |
| ResNet-18 | Single model | 95.07% | 77.16% |
| | MOE method | 94.46% | 77.22% |
| | Fast ensemble | 94.80% | 77.32% |
| | Snapshot ensemble | 95.72% | 79.40% |
| | Div² | - | 79.12%* |
| | SANE | **96.04%** | **81.10%** |
| ResNet-32 | Single model | 93.17% | 70.29% |
| | MOE method | 92.78% | 72.02% |
| | Fast ensemble | 92.11% | 71.56% |
| | Snapshot ensemble | 94.21% | 74.51% |
| | SANE | **94.92%** | **76.10%** |

**Base models and hyper-parameters.** We use MLP, whose implementation is in Appendix A.1.1, as the base model. For fair comparison, we keep the same hyper-parameters of the base models for random ensemble and our method.

### 4.1.2 IMAGE DATASETS

***CIFAR.*** CIFAR-10 and CIFAR-100 are the standard image classification benchmarks (Krizhevsky et al., 2009), containing 50,000 and 10,000 samples, labeled as 10 and 100 classes, respectively. From the training set, we randomly select 2,500 samples to compose the validation set and apply the standard data augmentation detailed in He et al. (2016b).

**Baselines.** We compare SANE with several state-of-the-art ensemble algorithms applied in image datasets, including **Snapshot Ensemble** (Huang et al., 2017), **Fast Ensemble** (Garipov et al., 2018), Multi-Model Ensemble via Adversarial Learning (**MEAL**) (Shen et al., 2019), **Diversified Ensemble** (Zhang et al., 2020), and Diverse ensemble evolution (**Div²**) (Zhou et al., 2018). Additionally, we also implement mixture-of-experts (**MOE method**) with reference to Shazeer et al. (2017) for a soft gating network, see the Appendix A.1.3 for implementation details.

**Base models and hyper-parameters.** We use VGGNet-19 (Simonyan & Zisserman, 2014), ResNet-18, and ResNet-32 (He et al., 2016a) as the backbone networks for the CIFAR datasets, and follow the standard hyper-parameters.

### 4.2 EVALUATION ANALYSIS

We assess the performance of SANE and baselines on tabular and image datasets. The results are shown in Table 1 and Table 2.

**Tabular dataset results.** SANE outperforms all the tree-based methods and the state-of-the-art TabNet method on both datasets (**RQ1**) which illustrates the superior performance of our method in tabular datasets with various scales.

**Image dataset results.** For the CIFAR dataset, experiments are conducted on several backbone networks. We can see that SANE is consistently superior to all the baselines (**RQ1**). Among all methods using VGGNet-19 on CIFAR-100 data, SANE outperforms the second one (Diverisfied Ensemble) by 2.70%. That is to say, specialization-driven SANE is better than implicitly and explicitly diversity-driven methods. Recall that Div² assigns a subset of samples to the base models with static expertise. By comparing its result with that of SANE, we can see that SANE with active

specialization on specific regions in latent space works better (**RQ2**). Additionally, SANE brings higher improvement on CIFAR-100 than CIFAR-10, which reveals that it may perform better when the task is relatively more difficult. The MOE-based approach performs worse than a single model in most cases. We speculate the reason that MOE uses ensemble loss to update base models and the gating network, resulting in diverse base models while aggregation does not achieve the effect of specialized aggregation. This result can also correspond to our ablation setting ⑦ of SANE only using $\mathcal{L}_e$, in Section 4.3. This suggests that $\mathcal{L}_e$ is not the main component that makes SANE work. For comparison with papers without publicly available code, here we only present results with the number of base models as $K = 3$. Results for other $K$ values can be found in Appendix A.2.

## 4.3 ABLATION STUDIES

To investigate the implications of different components in our model, we design ablation experiments with 8 different settings, which are performed in CIFAR-100 using VGGNet-19. Recall that, we adopt three losses $\mathcal{L}_b$, $\mathcal{L}_c$ and $\mathcal{L}_e$ that effectively serve our ensemble mechanism, thus we first conduct ablation study over them. Moreover, as mentioned in Section 3.2.3, we also empirically demonstrate the effect of detaching the update of base model parameters $\{\theta_k\}_{k=1}^K$ from $\mathcal{L}_e$'s computation graph (referred to as detach, otherwise attach) on the ensemble performance.

The results are presented in Figure 2, from which it is clear that each loss plays a part (**RQ3**). It seems that $\mathcal{L}_b$ in Eq. (2) is the most necessary one, as the two settings without it perform worse than the single model. After discarding $\mathcal{L}_c$ and $\mathcal{L}_e$, the performance is respectively degraded by 1.04% and 0.33% while still outperforming the baselines.

Additionally, we can compare the performance the base model detachment operation in two different settings (with or without $\mathcal{L}_c$), as ①④ and ③⑥ in Figure 2. The ensemble loss $\mathcal{L}_e$ naturally assigns the weights based on the model's prediction confidence over the true label class in back-propagation. From the results, the detachment affects the performance in opposite ways for the two different settings. In setting ④, the current specialization of the base model is encouraged by $\mathcal{L}_c$ and $\mathcal{L}_e$, but no better final results are achieved than that in setting ①. In contrast, in the absence of $\mathcal{L}_c$, the loss of $\mathcal{L}_e$ in the ensemble makes the base model more specialized when trained under setting ③, which brings some performance improvement. This suggests that ensembles can benefit from specialized base models. Notably, in addition to assigning weights based on the model's current specialization, $\mathcal{L}_c$ encourages the model to explore potential specializations. By comparing the results of $\mathcal{L}_e$-driven model specialization (setting ③) with that of our adaptive specialization (setting ①), it illustrates that our approach yields superior performance (**RQ2**).

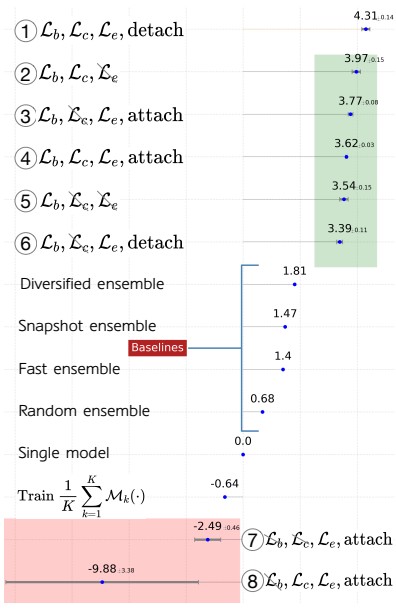

**Figure 2:** Ablation studies on CIFAR-100 using VGGNet-19. The values represent the performance difference, average over 3 runs, of each method relative to the single base model.

## 4.4 FURTHER ANALYSIS

**Analysis of base models' specialization** By design, base models need to specialize in different regions. It is worth noting that the regions (distributions) in which a model specializes do not correspond to data classes. Here we only provide an intuitive visualization to answer **RQ4**. Since inter class distribution differences are generally more significant than those of intra class, we attempt to analyze base models' specializations on CIFAR-100 superclasses, by finding *expressed specialists of samples. Ideally, samples of the same class would be dominantly specialized by one model.* Figure 3 shows the amount of samples (proportional to the connection width) for which the model specialized in that class. The 3 base models (A, B, and C illustrated on the right side) trained by SANE show divergent specialization on different data classes (on the left side).

**The number of parameters added for ensemble network.** The ensemble network can be viewed as a model-agnostic framework. For different tasks, only the feature dimension and the number of base models will affect the number of parameters. When using ResNet-18 as the base model, the ensemble network uses only 4.0% of the number of parameters of the single model to control 3-model ensemble, which improves the performance of CIFAR-100 by 5.1%.

## 5 RELATED WORK

Ensemble method has been proven effective for practical machine learning scenarios (Hansen & Salamon, 1990; Schapire, 1990; Zhou et al., 2002). There are mainly two phases in ensemble methods, i.e., base model training and model aggregation, and many works are studying either or both of them with emphasizing diversity. In the first phase, for example, bagging (Breiman, 1996) proposed to randomly select the subsets from the whole training data for training various base models. However, literature (Lee et al., 2015) shows that the base models in bagging cannot make full use of data yet suppresses the performance of deep neural networks. The similar issue also occurs in other sample-model matching methods such as Zhou et al. (2018); Liu et al. (2020). Other approaches relying on random initialization (Zhang et al., 2020; Lakshminarayanan et al., 2016) may derive similar base models lacking diversity, as shown in this paper. Our method is in contrast to Bayesian neural networks which estimate uncertainty through prior assumption (Gal & Ghahramani, 2016) and rely on randomization.

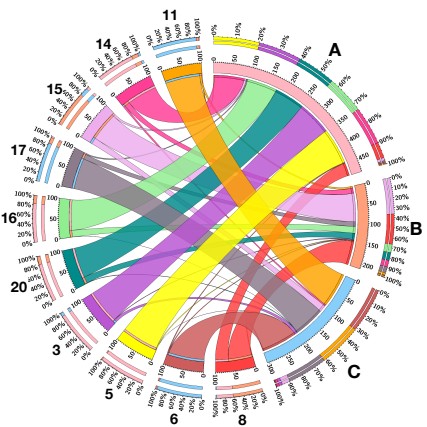

**Figure 3:** Proportion of samples specialized by the 3 VGGNet-19 models (A/B/C) trained with SANE for the 10 CIFAR-100 superclasses (labeled as numbers).

There also exist some other issues in the second phase of ensemble learning or the joint optimization methods. Most works adopt averaging over the outputs from all the base models, which has not demonstrated promising performance after ensemble operation (Lee et al., 2015). Some works combine the two phases and propose to directly learn all the baseline models under the loss upon ensemble (Zhang et al., 2020; Zhou et al., 2018). However, Lee et al. (2015) has pointed out that ensemble-aware loss through averaging base model predictions incorporates instability.

Other methods incorporate boosting (Schapire, 1990; Freund, 1995; Moghimi et al., 2016) which focuses on weighted prediction combination through a sequentially training paradigm for deriving a series of base models where each of them is optimized based on the condition derived from the previous ones. Other sequential optimization methods also include snapshot ensemble (Huang et al., 2017), however, either boosting or snapshot ensemble is hard to parallelize and can lead to long training time when applied to deep neural networks (Zhou et al., 2018).

Our work is also related to mixture-of-experts methods, which have been proposed to tackle the problems of combining some specialized experts to achieve promising ensemble performance (Jacobs et al., 1991; Jordan & Jacobs, 1994; Shazeer et al., 2017). However, the existing MOE methods neither consider local patterns explicitly, nor pay enough attention to effective model aggregation. Our method learns the specialized base models and specialization-aware model aggregation simultaneously, which can achieve superior performance as demonstrated in the experiments.

## 6 CONCLUSION

Real-world data is often generated by some complex distribution, which can be approximated by a composition of several simpler distributions. Existing ensemble methods that diversify base models are not able to explicitly enforce model specialization. Therefore, this paper proposes an end-to-end ensemble learning method that actively enforces model specialization, where base models are trained to specialize in sub-regions of a latent space representing the simple distribution composition, and aggregated based on their specialty. Experiments on both image datasets and tabular datasets demonstrate the superior performance of our proposed method.

ETHICS STATEMENT

Ensemble learning is a general method for machine learning. To the best of our knowledge, there are no intentional ethical issues created by the proposed method.

REPRODUCIBILITY STATEMENT

The datasets used in this paper are all publicly available, and all implementation details are introduced either in the main file or in the appendix. In addition, we will release the source code of all the experiments upon paper acceptance.

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

# A   APPENDIX

## A.1   IMPLEMENTATION DETAILS

In this section, we present the details of the base model design, hyper-parameter setup, and implementation of the MOE method. For all tasks, we first tune the hyper-parameters of the base model and then fix them. The learning rate of the base model is kept by hyper-parameter tuning of the base model, thus, the loss weight $s_b$ in Eq. (5) is set to 1. For the ensemble network, we tune its learning rate, $s_c$, and $s_e$. Additionally, the embedding dimension of specialty anchors is also a hyper-parameter. Table 3 lists the hyper-parameters to tune and their search space. For all tasks, we run a hyper-parameter tuning with 100 search steps. In the ensemble network, the projections $\mathcal{Q}, \mathcal{K}$, and $\mathcal{V}$ and the scale-transfer layer are implemented as fully connected layers, which are the same for both image and tabular datasets. The temperature parameters $\tau_b$ and $\tau_e$ are set as 0.12 and 0.01, respectively.

**Table 3:** Hyper-parameters we tune for the ensemble network. We search learning rates using log uniform.

| Hyper-parameter | Search Range |
|---|---|
| learning rate of ensemble network | [0.0001,0.1] |
| loss weight $s_c$ | [1,2,3,4,5] |
| loss weight $s_e$ | [1,2,3,4,5] |
| embedding dimension of specialty anchors $\{\mathbf{a}_k\}_{k=1}^K$ | [25,50,75,100] |

From the results, SANE works on larger datasets using larger models as well and consistently outperforms baselines.

### A.1.1   IMPLEMENTATION DETAILS FOR TABULAR DATASETS

**Base models.** For tabular datasets, we use an MLP as the base model. For each layer of the MLP, the fully connected layer is followed by a batch normalization layer and a residual connection, using the gated linear unit (GLU) as the activation function. The feature encoder $\mathcal{F}$ in the ensemble network is implemented as MLP of the same structure as the base model, and we take the output of the last fully connected layer as the feature embedding. For the Higgs Boson dataset, we follow the network structure used by Baldi et al. (2014), which is a 5-layer MLP with 300 hidden units in each layer. For Blastchar, we use a 2-layer MLP with 32 hidden units in each layer.

**Hyper-parameter setup.** For the Higgs Boson dataset, we follow most of the hyper-parameters in Baldi et al. (2014), except that the learning rate is modified to 0.01. For Blastchar, we use Adam with a learning rate of 0.001 and a weight decay of 0.001.

### A.1.2   IMPLEMENTATION DETAILS FOR IMAGE DATASETS

**Base models.** For image datasets, we use several backbones, including VGGNet-19, ResNet-18, and ResNet-32. Implementations of these backbones are standard. We use a pre-trained backbone corresponding to the base model and freeze it as the feature encoder $\mathcal{F}$ in the ensemble network. The pre-trained VGGNet-19 and ResNet-18 are taken from TorchVision, an open-source machine vision package for Torch. Since there is no available pre-trained ResNet-32 available in TorchVision, we use VGGNet-19 as a replacement. All of these pre-trained model outputs a 1,000-dimensional feature embedding.

**Hyper-parameter setup.** On the CIFAR datasets, the standard parameters were applied for these commonly used backbones. For VGGNet-19 and ResNet-18, we follow the hyper-parameters used in DeVries & Taylor (2017). For ResNet-32, we use the same hyper-parameters from Chen et al. (2020). To be specific, for VGGNet-19 and Resnet-18, the initial learning rate was 0.1, and the learning rate was divided by 5 for the 60th, 120th, and 160th epochs, with 200 epochs of training, a batch size of 128, a weight decay of 5e-4, and a Nesterov momentum of 0.9. For ResNet32, we trained for 300 epochs, with all other parameters unchanged, while the learning rate was divided by 10 at the 150th and 225th epochs.

### A.1.3 IMPLEMENTATION DETAILS OF MIXTURE-OF-EXPERT METHOD

In similar fashion to our framework, MOE consists of a set of expert networks $\{E_i\}_{i=1}^n$ and a gating network $G$. For $\mathbf{x}_n$, the output of MOE is $\sum_{i=1}^n G(\mathbf{x}_n)_i E_i(\mathbf{x}_n)$. In Shazeer et al. (2017), the gating network is realized as $G(x) = \text{Softmax}(x \cdot W_g)$ where $W_g$ is a trainable weight matrix. The training of expert networks and the gating network is done by back-propagation of cross entropy loss between its outputs and ground truth, i.e., analogously to our $\mathcal{L}_e$ in Eq. (4). To present a fair comparison with ours, we further strength the gating network as $G(x) = \text{Softmax}(\text{W}(\mathcal{F}(x)))$, where $\mathcal{F}$ is the feature encoder and W is an MLP with 2 hidden layers.

### A.2 NUMBER OF BASE MODELS $K$

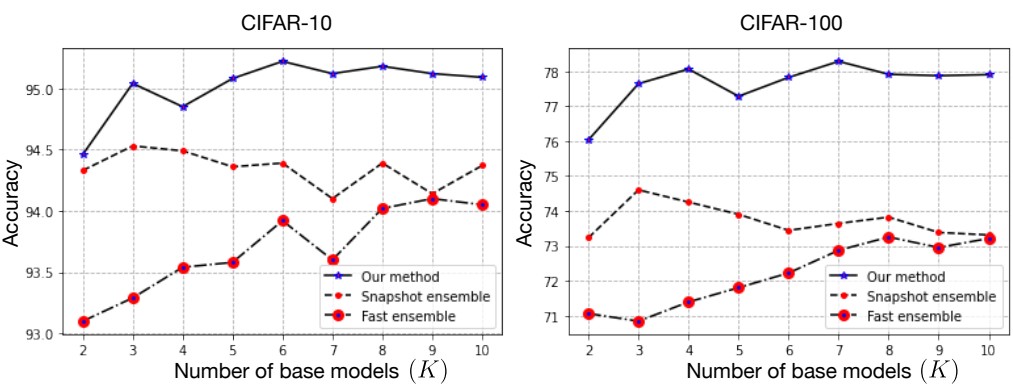

**Figure 4:** Accuracy comparison of methods with different number of base models $K$ on CIFAR-10 and CIFAR-100.

Figure 4 presents the accuracy of CIFAR-10/100 with different number of base models $K$. Here we compare with snapshot ensembles and fast ensembles, which perform worse than our method at any $K$ value. The best performance for CIFAR-10 and -100 is obtained when $K$ is equal to 6 and 7, respectively, as seen in the figures.

## B EXPERIMENTAL RESULTS ON TINY IMAGENET

In addition to the CIFAR dataset, we also conduct experiments on Tiny ImageNet with more complex data distribution.

**Base models.** While previous papers focused on the CIFAR dataset, Snapshot Ensemble conducted experiments on Tiny ImageNet using ResNet-110 as the backbone network, which we followed. In addition, experiment on EfficientNet-B0, a promising choice on the dataset, is performed using the same hyper-parameters of (Luo et al., 2019).

We compared with Random Ensemble, Snapshot Ensemble, and Fast Ensemble, and results are as follows.

**Table 4:** The comparison experiments on Tiny ImageNet using different backbones, with ensemble size of 3.

|  | EfficientNet-B0 | ResNet-110 |
|---|---|---|
| Single model | 59.34 | 51.18 |
| Fast ensemble | 60.66 | 53.70 |
| Snapshot ensemble | 64.59 | 54.03 |
| Random ensemble | 64.71 | 56.32 |
| SANE | **65.19** | **57.01** |

## C    RESULTS ON THE CIFAR DATASET COMPARED TO THE SUPPLEMENTAL BASELINES

Apart from baselines listed in Section 4.1.2, Stochastic Weight Averaging (Izmailov et al., 2018) and Hyper-parameter Ensemble (Wenzel et al., 2020) are also STOA methods in ensemble learning. Due to space limitation, we present the results of the comparison experiments in this section.

For Stochastic Weight Averaging, we conduct its experiments on CIFAR-10/100 with VGGNet-19 using settings consistent with our experiments to ensure fair comparison. For the Hyper-parameter Ensemble, we re-experiment SANE on ResNet-20 to compare with the results presented in their paper. The comparison results are as follows:

**Table 5:** Complete comparison results on CIFAR-10 and CIFAR-100 with $K = 3$. The $*$ indicates result is taken from the original paper, while the $\S$ represents from Zhang et al. (2020).

| Backbone | Method | CIFAR-10 | CIFAR-100 |
|---|---|---|---|
| | | Accuracy (top-1) | |
| VGGNet-19 | Single model | 93.21% | 73.13% |
| | Random ensemble | 93.80%$\S$ | 73.81%$\S$ |
| | MOE method | 92.87% | 70.46% |
| | Stochastic Weight Averaging | 93.42% | 72.84% |
| | Fast ensemble | 93.29% | 74.53% |
| | Snapshot ensemble | 94.53% | 74.60% |
| | MEAL | 94.45%$*$ | - |
| | Diversified ensemble | 94.91%$*$ | 74.94%$*$ |
| | SANE | **95.04%** | **77.64%** |

**Table 6:** The comparison experiments on CIFAR-10/100 using ResNet-20 between SANE and Hyper-parameter Ensemble marked in green, with $K = 4$. The results of Hyper-parameter Ensemble are taken from the original paper (Wenzel et al., 2020). The values in parentheses represent the performance improvement brought by the ensemble compared to the single model results.

| Backbone | ResNet-20 | |
|---|---|---|
| Method | Hyper-parameter Ensemble | SANE |
| CIFAR-10 | 94.00% (+ 1.30%) | **94.60% (+ 2.18%)** |
| CIFAR-100 | 74.20% (+ 6.00%) | **74.82% (+ 6.82%)** |

From the results, SANE outperforms Stochastic Weight Averaging and Hyper-parameter Ensemble in a fair comparison setting.

## D    ENSEMBLE OF DIVERSE BASE MODELS

In previous experiments, we focus on the ensemble of base models with the same architecture, since evaluating the effectiveness and generalization of the framework on different data types is the main target. That is, the only type of randomness that exists in SANE is the different initialization of base models. Gontijo-Lopes et al. (2021) claimed that models across different hyper-parameters, architectures, frameworks, and datasets could be specialized for sub-domains of data, leading to higher ensemble performance. It confirmed the power of specialization in ensemble learning, although they propose to implicitly achieve specialization rather than explicitly as in our method.

To verify the validity of pursuing specialization explicitly, we compare the performance of specialization-aware ensemble on diverse base models with that of specialization achieved by implicit manner, i.e., a random ensemble of diverse architectures. Moreover, we measure and visualize the specialization of the models trained by the two methods respectively.

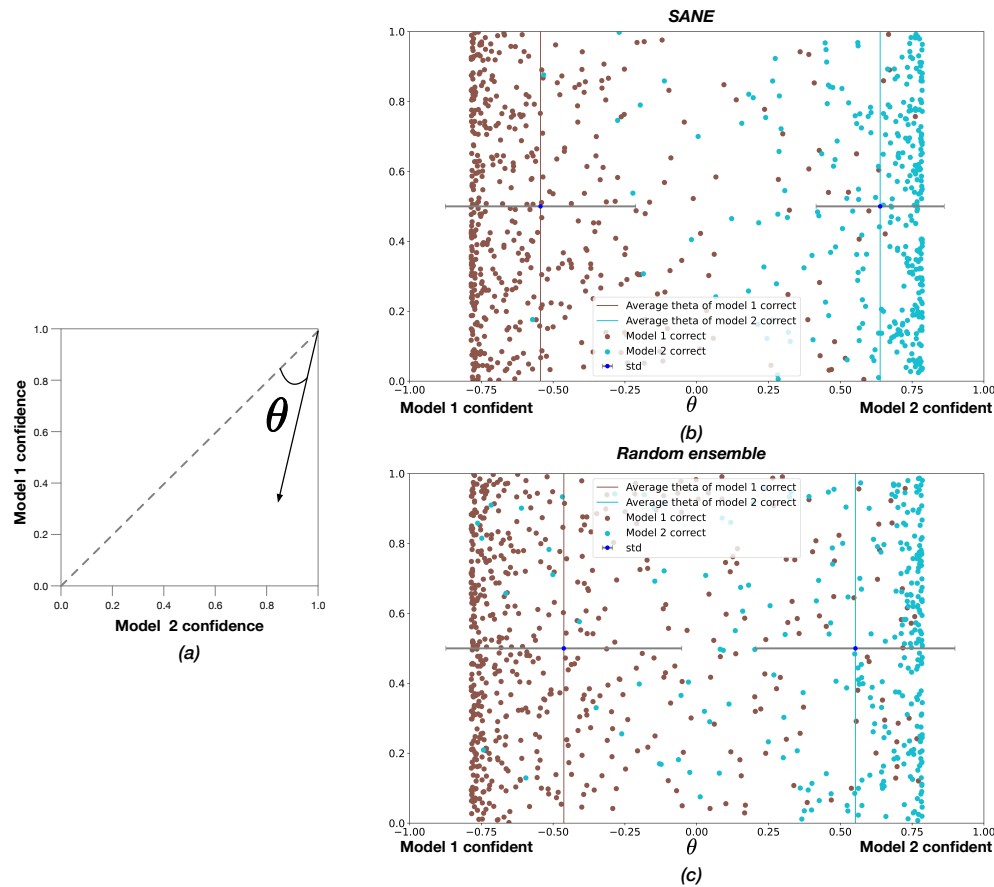

**Figure 5:** (a) the confidence-confidence plot with $\theta$ as the specialization measurement defined in Gontijo-Lopes et al. (2021). (b) and (c) are specialization measurement of two base models trained by SANE and independently trained when only model 1/2 correct, respectively.

**Base models and hyper-parameters.** We use three different architectures, i.e., VGGNet-19, ResNet-18, and DenseNet-40. All the hyper-parameter setup is the same as in Appendix A.1.2.

From the results, we fing that the SANE-trained ensemble (78.37%) outperform the random ensemble (76.83%) by 1.54%. It indicates that, under the same network architectures and model size setting, even though architectural diversity can benefit the ensemble by bringing implicit specialization, explicitly encouraging specialization will benefit the ensemble further.

In Gontijo-Lopes et al. (2021), they define a specialization metric $\theta$ as the angular distance $\theta$ in the confidence-confidence graph, as shown in Figure 5(a), and visualize it. Inspired by that, we show the specialization $\theta$ of the two base models (ResNet-18 and DenseNet-40) in Figure 5(b) when only model 1/2 makes a correct prediction, respectively. We compare with the random ensemble with architecture and initialization diversity and no explicit specialization guidance, and the results are shown in Figure 5(c). In the figure, we label the average $\theta$ in both cases. As can be seen, while both ensembles present that 'when only one model makes the correct prediction, it tends to be more confident', SANE's base model exhibits a higher level of specialization compared to the baseline.

Additionally, though effective, introducing randomness in hyper-parameters, architectures, frameworks, and datasets will require a lot of expert knowledge and manual efforts. In contrast, our method is a generic framework with the ability to learn to achieve ensemble and explicitly enforce specialization.

