# OpenReview forum: "SANE: Specialization-Aware Neural Network Ensemble"
_ICLR.cc/2022/Conference — ICLR 2022 Submitted_

### Official Review · Reviewer_y2rX · 2021-10-27

**Correctness:** 4
**Technical Novelty And Significance:** 2
**Empirical Novelty And Significance:** 2
**Recommendation:** 6
**Confidence:** 4

**Main Review:**

* Strengths:
1. The paper is well-written and its idea is easy to follow
2. This paper presents a new perspective - specialization - to enhance the ensemble learning. By the proposed approach, the problems of existing strategies can be solved.

* Weakness:
1. Although the analysis perspective is new, but the technique used in this paper are current existing one. The core for distributing samples in the latent space is based on the attention machanism. According to the description in the manuscript, this part is a direct usage of the one in transformer. This leads to my concern on the novelty of this paper.
2. I think the number of specialization anchors is one of the most important hyper-parameter of the proposed approach. However, the analysis on this hyper-param. is very limited. According the appendix, the maximum of this param is only 10 in the experiments. I am concerned about if the setting is still valid on more complex data distribution, such as the common use one, ImageNet dataset. To make the conclusion more convincing, I think the author should provide analysis on this hyper-param. or experiments on large-scale dataset.

**Summary Of The Paper:**

This paper presents a SANE model to improve the ensemble learning from the perspective of model specialization. In particular, it first gives the analysis on existing ensemble strategies, i.e., random and diversity-driven, via conducting experiments on synthetic data. By this way, it points out the weakness of these strategies lies in the lack of specialty. Motivated by this, this paper presents the specialization-aware method to improve the ensemble learning. Specifically, it introduces anchor points in the latent space for guiding the model learning towards specialty. To derive the correlation between samples and anchors, it utilizes the transformer-like attention mechanism for learning the weights of base models. This paper demonstrates its effectiveness on several image and tabular benchmarks.

**Summary Of The Review:**

In summary, I think the idea for encouraging specialization in the model ensemble is rational and novel. But I am conerned about the novelty of the specific technique adopted in this paper as well as the insufficient experiments.

---

> ### Author Response · Authors · 2021-11-19
> **Response to Reviewer y2rX**
>
> Thanks for your constructive suggestions and comments, which are very helpful for us to improve this paper. Next, we would like to respond to each of the concerns raised in your comments.
>
> > **Q1. "Although the analysis perspective is new, but the technique used in this paper are current existing one."**
>
> We agree that the implementation of the ensemble net is not new. The technical details of the ensemble network may be a further challenge, as for complex designs which bring higher computational costs, raising the trade-off between effectiveness and efficiency. However, we would like to emphasize that the main contribution of this work is to propose this ensemble learning framework with performance-aware capabilities, which is a novel proof of concept. We believe this new framework can help to inspire other researchers to design ensemble learning methods from a new (and maybe more promising) perspective.
>
> > **Q2. "the maximum of the number of base models $K$ is only 10 in the experiments"**
>
> The computational cost of ensemble learning makes it difficult to ensemble larger model sizes, and therefore, 10 is also a common choice in the previous literature (e.g., a maximum of 6 base models in [1]). Moreover, from the experimental results, we found that the benefits of large ensemble size saturate at ensemble sizes of 6 and 7 in most of the experiments. Therefore, 10 may be a reasonable choice in practice. But we do agree with you that $K$ is an important hyperparameter and should be carefully tuned, especially in complicated scenarios, e.g., the ImageNet dataset as mentioned in your following question.
>
> > **Q3. I am concerned about if the setting is still valid on more complex data distribution, such as the common use one, ImageNet dataset.**
>
> Thank you for your comments. Following this suggestion, we added experiments on Tiny ImageNet with more complex data distribution, with results presented in Table 1 in the "Reply to All" and also in Table 4 of the revised manuscript.
>
> Reference:\
> [1] Zhang, Shaofeng, Meng Liu, and Junchi Yan. "The diversified ensemble neural network." Advances in Neural Information Processing Systems 33 (2020).

---

### Official Review · Reviewer_G9Xj · 2021-11-02

**Correctness:** 3
**Technical Novelty And Significance:** 2
**Empirical Novelty And Significance:** 2
**Recommendation:** 5
**Confidence:** 3

**Main Review:**

Explicit incorporation of the model diversity through sub-region.
No theoretical justification to support the claim that it will give a better ensemble model.

**Summary Of The Paper:**

This paper learns an ensemble of models where each of the base models has a specialization in some latent sub-region.

**Summary Of The Review:**

The use of specialized models on sub-regions can be very interesting for shallow models but for complex models such as deep networks, it is not convincing that the models specialized in sub-regions can bring any change.

How the authors found their results different than predicting a sample using only the subregions models by maintaining a mapping for a test sample to a sub-region model.


The results show the only marginal differences.

---

> ### Author Response · Authors · 2021-11-19
> **Response to Reviewer G9Xj (part 1)**
>
> Thanks for your constructive suggestions and comments, which are very helpful for us to improve this paper. Next, we would like to respond to each of the concerns raised in your comments.
>
> > **Q1. "Explicit incorporation of the model diversity through sub-region. No theoretical justification to support the claim that it will give a better ensemble model."**
>
> We first provide a theoretical justification (following [1]) of why sample-wise specialization promotes ensemble accuracy and then explain the intuition of employing model specification through sub-region.
>
> #### 1.1 Theoretical justification
>
> For voting-based ensemble in classification task, we can define an aggregated predictor $M$. Given $x$, the probability of $M$ predict its class label as $c \in \{1, \dots, C\}$ is $Q(c|x)$. Thus, for $x$, the probability of the aggregated predictor predict correctly is $\sum_c Q(c|x)P(c|x),$ where $P(c|x)$ is the probability of input $x$ corresponds to class $c$.
>
> The overall correctness probability is $$r = \int \sum_c Q(c|x)P(c|x) dx.$$ For any $Q(c|x)$, $$\sum_c Q(c|x)P(c|x) \leq \max_{c}P(c|x),$$ with equality only if $Q(c|x) = \mathbf{I}[P(c|x) = \max_{i}P(i|x)]$.
>
> If we have a bayes predictor $M^{\ast}(x) = \arg\max\limits_{i}P(i|x)$, it can obtain the upper bound of classification correctness: $$r = \int \max_{i}P(i|x) dx.$$ We can call $M$ order-correct if $\arg\max\limits_{i}Q(i|x) = \arg\max\limits_{j}P(j|x).$ Thus, with $M(x) = \arg\max\limits_{c}Q(c|x)$, the overall correctness probability can be written as $$r = \int_{x\in \mathcal{E}} \max_{i}P(i|x) dx + \int_{x\notin \mathcal{E}} \left[ \sum_i \mathbf{I}(M(x) = i) P(i|x) \right] dx,$$ where $\mathcal{E}$ is the subset of dataset where $M$ is order-correct.
>
> Therefore, obtaining higher $r$ is equal to expanding $\mathcal{E}$. In contrast to giving each base learner with the same voting weight, we attempt to give order-correct base learners more voting weight thus enforcing $M$ to have a higher correct-order rate.
>
> #### 1.2 Local specialization motivation
> For many tasks, it is difficult to train a model to be excellent over the whole sample space. Therefore, partitioning the entire sample space into multiple local regions and training each base model to be a specialist in one region (while keeping it aware of the whole sample space) is a feasible solution, similar to the intuition of many large problems, such as the local low-rank matrix decomposition in [2].
>
> > **Q2. "The use of specialized models on sub-regions can be very interesting for shallow models but for complex models such as deep networks (big models), it is not convincing that the models specialized in sub-regions can bring any change."**
>
> Regarding your question about the performance of SANE on large models, we added experiments on Tiny ImageNet using larger models, with results presented in Table 1 in the "Reply to All". From the results, SANE also works for complex models such as EfficientNet.
>
> > **Q3. "How the authors found their results different than predicting a sample using only the subregions models by maintaining a mapping for a test sample to a sub-region model."**
>
> Mixture-of-expert is a popular method to maintain a sample-to-model mapping, except that a sample will be assigned to a subset of sub-region models. We have compared SANE with MOE and found that SANE outperformed MOE significantly.
>
> If our understanding is correct, you actually mean maintaining a mapping that assigns each one sample to only one base model. To address this concern, we experimented with a modified version of MOE in which the gated network generates a one-hot weight vector using Gumbel-Softmax with a reparameterization trick. We found that this hard assignment method makes the performance degrade significantly. The main reason is that combining the predictions from multiple base models can achieve better bias-variance tradeoff as demonstrated in many prior works on ensemble learning, leading to improved generalization performance in test sets. Therefore, we propose to combine multiple specialized base models through the learned weights rather than simply assigning the sample to a sub-region model.

---

> ### Author Response · Authors · 2021-11-19
> **Response to Reviewer G9Xj (part 2)**
>
> > **Q4. "The results show the only marginal differences"**
>
> From Table 2 in our paper, we can see that SANE consistently outperforms all baselines. Among all methods using VGGNet-19 on CIFAR-100, SANE outperforms the second-place method [4] by 2.70%, bringing nearly a 2-fold improvement over other STOA ensemble methods. Although, the results suggest SANE performs better on harder tasks (e.g., CIFAR-100) than easier tasks (e.g., CIFAR-10). However, in practice, it is reasonable to adopt ensembles to enhance the performance of difficult tasks, for which single models have limited solution capability.
>
> References:\
> [1] Breiman L. Bagging predictors[J]. Machine learning, 1996, 24(2): 123-140.\
> [2] Lee, Joonseok, et al. "LLORMA: Local low-rank matrix approximation." (2016).\
> [3] Lee, Stefan, et al. "Why M heads are better than one: Training a diverse ensemble of deep networks." arXiv preprint arXiv:1511.06314 (2015).\
> [4] Zhang, Shaofeng, Meng Liu, and Junchi Yan. "The diversified ensemble neural network." Advances in Neural Information Processing Systems 33 (2020).

---

### Official Review · Reviewer_etDH · 2021-11-03

**Correctness:** 2
**Technical Novelty And Significance:** 3
**Empirical Novelty And Significance:** 1
**Recommendation:** 6
**Confidence:** 5

**Main Review:**

Major comments

Motivation: the paper claims it is “intuitive to divide the complex model learning into training several simpler models.” However, such an approach has thus far not been shown to yield high performance [1]. In my opinion, the paper should justify why the approach presented fixes the issues that lead to [1] not finding high-performance diverse models.

Correctness: the paper claims it shows that  “without explicitly enforcing specialization, pursuing diversity may not be enough to achieve satisfactory ensemble performance,” but it only investigates a very narrow set of tasks. In particular, concurrent work [2] shows that with different enough models, one can achieve specialization (and high ensemble performance) without explicitly training for it. This points to the fact that the paper’s baselines of ensembling randomly-reinitialized models is not a good enough baseline for this claim. The paper also claims that “without explicitly enforcing model specialization, pursuing model diversity does not provide guarantees of each model’s specialization on different simple distribution in the composition”, but it is not shown what kinds of guarantees the specialization-enforcement losses presented yield.

Experiments: the paper mainly focuses on tabular data and small image datasets, with small models (the largest used is ResNet 32). They compare their method only with randomly-reinitialized ensembles, and other diversity-inducing methods, never even ensembling two networks together, or using SOTA ensemble techniques such as weight-averaging/EMA, or hyper-parameter ensembling. This makes it difficult to see how the results presented actually compare to realistic settings.



[1] http://www.gatsby.ucl.ac.uk/~balaji/why_arent_bootstrapped_neural_networks_better.pdf
[2] https://openreview.net/pdf?id=BK-4qbGgIE3

Minor comments

“the proposed specialization-aware neural network ensemble method can achieve superior performance compared with the state-of-the-art ensemble methods in various prediction tasks.” SOTA methods include weight-averaging and hyper-parameter ensembling, so this sentence strikes me as incorrect.

“However, overvaluing diversity may hurt the ensemble effectiveness and the
trade-off between the performance and diversity of base models still remains an open problem.”
Citation needed?

It’s unclear what the message of Fig 3 is. It strikes me as uninformative.

Many claims of what there “should” be in a design, without a lot of motivation, or ablation
“there should be an evolutionary process in the specialization of the model. The specialty regions of the models should be explored and learned in an adaptive manner”

Fig 2 should include error bars (multiple runs of each experiment), especially since doing these on small datasets like cifar-100.


**Summary Of The Paper:**

This paper proposes a method to train models that specialize in subsets of the data distribution by weighting the training loss based on how close example embeddings are to a learned “anchor” embedding. It also proposes to improve ensembling by combining multiple “specialized” models.


**Summary Of The Review:**

In general, I think finding methods to encourage specialization in models (while still achieving high accuracies) is a very important direction to pursue. The method presented of how to train specialized models (using anchors) is intuitive, and might be useful if demonstrated that it works. Unfortunately, it’s not possible to assert this from the limited baselines/experiments presented.

---

> ### Author Response · Authors · 2021-11-19
> **Response to Reviewer etDH (part 1)**
>
> Thanks for your constructive suggestions and comments, which are very helpful for us to improve this paper. Next, we would like to respond to each of the concerns raised in your comments.
>
> > **Q1. Why SANE "fixes the issues that lead to [1] not finding high-performance diverse models."?**
>
> The main drawback of the bagging method in ensemble learning is that base learners only see part of the samples, which is shown to suppress the performance of deep neural networks ([1]; [2] mentioned in your review). In contrast, the base models of SANE are aware of the entire training data. That is one of the reasons why SANE can obtain promising performance.
>
> Previous ensemble learning algorithms attempt to assign different samples/features to different base learners depending on their performance, i.e., the bagging approach. Since base learners only see part of the samples, they may fail to be aware of the whole data distribution to design optimal specialized base models.
>
> Different from the prior works, the base models of SANE are aware of the entire training data distribution during training, which has higher potential to learn optimal specialized base models. Note that, the key difference to simply training over the entire training set is that, different base models in SANE will be assigned different learning weights on different samples according to the speciality they reveal during training.
>
> > **Q2. The claim of “without explicitly enforcing specialization, pursuing diversity may not be enough to achieve satisfactory ensemble performance” is investigated on narrow set of tasks.**
>
> By exploration on synthetic data, we provide an intuitive understanding of how diversity and specialization work differently. In addition, we conduct comprehensive experiments on tabular/image datasets, and compared with various diversity-driven methods, SANE can achieve higher accuracy as indicated in Table 2. Both the case study on the synthetic dataset and the experiments on tabular/image datasets have demonstrated that only pursuing diversity as in many existing ensemble methods may not be enough to achieve satisfactory ensemble learning.
>
> > **Q3. "Concurrent work [2] shows that with different enough models, one can achieve specialization (and high ensemble performance) without explicitly training for it."**
>
> Thanks for pointing out this concurrent work, which also confirmed the power of specialization in ensemble learning (although they propose to implicitly achieve specialization rather than explicitly as in our method). However, specialization introduced implicitly by randomness requires huge manual efforts to ensure sufficient diversity. Furthermore, through experiments, we find that the enforcement of explicit specialization can further improve the performance of implicitly specialized ensembles. In addition, we visualize base models' specialization, demonstrating that explicit enforcement will lead to higher levels of specialization compared to the implicit methods.
>
> In the paper [3] that you mentioned, models across different hyper-parameters, architectures, frameworks, and datasets could be specialized for sub-domains of data, leading to higher ensemble performance. Though effective, introducing randomness in hyper-parameters, architectures, frameworks, and datasets will require a lot of expert knowledge and manual efforts. In contrast, our method is a generic framework with the ability of **learning to achieve ensemble and explicitly enforcing specialization**. To allay your concerns about how much benefit can be brought by explicitly enforcing specialization, we attached an experiment using three different architectures, namely VGGNet-19, ResNet-18, and DenseNet-40, which are trained using random ensemble and SANE, respectively. From the results, we found that the SANE-trained ensemble (78.37%) outperformed the random ensemble (76.83%) by 1.54%. This experiment indicates that, under the same network architecture and model size setting, even though architectural diversity can benefit the ensemble by bringing implicit specialization, explicitly encouraging specialization will benefit the ensemble further.
>
> Additionally, we follow the specialization measurement $\theta$ in [3] and present the comparison of SANE and random ensemble, which shows that explicitly enforcing specialization works and will be detailed in Q4 below.
>
> We are pleased to see that there is some work of interest on the topic of specialization-related ensembles. In Appendix D of the revised manuscript, we have added the experimental results on implicit and explicit guidance of specialization in ensemble learning and acknowledged this work.

---

> > ### Comment · Reviewer_etDH · 2021-11-30
> > **Response to authors**
> >
> > I thank the authors for their time and energy in spent in the paper and in the rebuttal, including extensive new experiments. I believe the paper is much improved, particularly with new experiments on image datasets and larger models, as well as the extra specialization analysis. It is interesting, for instance, to see that the three different architectures (VGGNet-19, ResNet-18, and DenseNet-40) not only benefit from SANE (+1.54%), but also display higher specialization measured by $\theta$. As i said in my original review, I think finding methods to encourage specialization in models (while still achieving high accuracies) is a very important direction to pursue, and I'm also pleased to see works pushing this direction.
> >
> > Originally, it was not clear to me that the method showcased was demonstrated to work well. With the updated paper, I am a bit more convinced of its merits. While I still think some claims seem exaggerated, and presentation could be improved (eg: Fig 3 could have a random ensemble companion for comparison), I think the authors' work during rebuttal has improved the paper. As such, I have raised my score to reflect this.

---

> > > ### Author Response · Authors · 2021-11-30
> > > **Response to Reviewer etDH**
> > >
> > > We would like to thank you for replying to our response. We appreciate your reviews, which helped us to enhance the paper to include more baselines, experiments on the larger dataset, and model specialization analysis. Regarding claims and presentation, we will refine them in the final version.

---

> ### Author Response · Authors · 2021-11-19
> **Response to Reviewer etDH (part 2)**
>
> > **Q4. The paper also claims that “without explicitly enforcing model specialization, pursuing model diversity does not provide guarantees of each model’s specialization on different simple distribution in the composition”, but it is not shown what kinds of guarantees the specialization-enforcement losses presented yield. / It’s unclear what the message of Fig 3 is. It strikes me as uninformative.**
>
> If our understanding is correct, these two concerns are related to the same question "Is there more significant specialization in the SANE-trained base models compared to other methods?"
>
> First, we will provide more clarification of the message conveyed in Figure 3. As described in Definition-2.1 and -2.2 in our paper, an ideal ensemble should contain *expressed specialist* for each sample. It contains two conditions: 1. at least one specialist has high prediction confidence; 2. other non-specialists produce low confidence predictions to avoid interfering with the predictions. Based on the idea, in Figure 3, we present the proportion of samples specialized by the 3 different base models trained with SANE for the 10 CIFAR-100 superclasses. It can be seen that in many superclasses, one model dominates as the *expressed specialist* on most of the samples in the same class.
>
>
> Moreover, to better evaluate the specialization achieved by various models and as per the complementary experiments in Q3, we additionally show the [specialization measurement $\theta$ for the two base models](https://postimg.cc/gXy3bk8c) as defined in [3] when only model 1/2 makes correct predictions as illustrated in Figure 5 of the revised manuscript, respectively. We then compare them to the ensemble with architectural and initialization diversity and no explicit specialization guidance. In the figure, we mark the average $\theta$. It can be seen that while both methods present that "when only one model makes a correct prediction, it tends to be more confident", SANE's base model presents a higher level of specialization compared to the baseline.
>
> > **Q5. "the paper mainly focuses on tabular data and small image datasets, with small models (the largest used is ResNet 32)."**
>
> Thanks for your suggestions. We added experiments on Tiny ImageNet using larger models, with results presented in Table 1 in the "Reply to All" and also in Table 4 of the revised manuscript.
>
> > **Q6. More existing baselines.**
>
> Thank you for providing additional baselines for comparison. Per your suggestion, we added baselines for the Stochastic Weight Averaging [4] and hyperparameter ensemble [5].
>
> For Stochastic Weight Averaging, we conduct its experiments on CIFAR-10/100 with VGGNet-19 using settings consistent with our experiments to ensure fair comparison. For the Hyperparameter Ensemble, we re-experiment SANE on ResNet-20 to compare with the results presented in their paper. The comparison results are as follows:
>
> | Backbone  | Method                                        | CIFAR-10 | CIFAR-100 |
> |-----------|-----------------------------------------------|----------|-----------|
> | VGGNet-19 | Stochastic Weight Averaging                   | 93.42    | 72.84     |
> |           | Fast Geometric Ensemble                       | 93.29    | 74.53     |
> |           | Multi-Model Ensemble via Adversarial Learning | 94.45    | /         |
> |           | Snapshot Ensemble                             | 94.53    | 74.60     |
> |           | Diversified Ensemble                          | 94.91    | 74.94     |
> |           | SANE                                          | **95.04**    | **77.64**     |
>
> | Backbone  | Method                   | CIFAR-10                                           | CIFAR-100                                          |
> |-----------|--------------------------|----------------------------------------------------|----------------------------------------------------|
> | ResNet-20 | Hyper-parameter Ensemble | 94.00% (+ 1.30% based on 92.70% single base model) | 74.20% (+ 6.00% based on 68.20% single base model) |
> |           | SANE                     | **94.60%** (+ 2.18% based on 92.42% single base model) | **74.82%** (+ 6.82% based on 68.00% single base model) |
>
> From the results, SANE outperforms Stochastic Weight Averaging and Hyper-parameter Ensemble in a fair comparison setting. Thank you for providing related baselines, we have added these experimental results and acknowledged these work in Appendix C in the revised version.
>
> > **Q7. “However, overvaluing diversity may hurt the ensemble effectiveness and the trade-off between the performance and diversity of base models still remains an open problem.” Citation needed?**
>
> Thank you for pointing out this issue. We have added the paper [6][7][8] as references for this argument. In these works, the authors claimed that the trade-off between ensemble diversity and individual accuracies is still an open problem.

---

> ### Author Response · Authors · 2021-11-19
> **Response to Reviewer etDH (part 3)**
>
> > **Q8. Motivation and ablation of “there should be an evolutionary process in the specialization of the model. The specialty regions of the models should be explored and learned in an adaptive manner”**
>
> ***Motivation:*** The motivation of adaptively learning the specialization is to avoid locking in some inappropriate specialty regions caused by initialization or learning procedure. It has also been discussed in [9] where they claim that samples should not be locked to certain models based on their initial performance, since model specialization evolves during training. We have revised the claim of motivation more clear in the revised manuscript.
>
> ***Ablation:*** The ablation by removing the specialization-forced loss in Eq. (3), which is the leader of evolution and exploration, is shown in Figure 2. From the results, the ensemble training evolutionarily adapted with the true confidence of base models makes a significant improvement in performance.
>
> > **Q9. Fig 2 should include error bars (multiple runs of each experiment), especially since doing these on small datasets like cifar-100.**
>
> Thank you for this suggestion. We updated Figure 2 in the revised manuscript.
>
> References:\
> [1] Lee, Stefan, et al. "Why M heads are better than one: Training a diverse ensemble of deep networks." arXiv preprint arXiv:1511.06314 (2015).\
> [2] http://www.gatsby.ucl.ac.uk/~balaji/why_arent_bootstrapped_neural_networks_better.pdf \
> [3] https://openreview.net/pdf?id=BK-4qbGgIE3 \
> [4] Izmailov, Pavel, et al. "Averaging weights leads to wider optima and better generalization." 34th Conference on Uncertainty in Artificial Intelligence 2018, UAI 2018. Association For Uncertainty in Artificial Intelligence (AUAI), 2018. \
> [5] Wenzel, Florian, et al. "Hyperparameter Ensembles for Robustness and Uncertainty Quantification."\
> [6] Rame, Alexandre, and Matthieu Cord. "DICE: Diversity in Deep Ensembles via Conditional Redundancy Adversarial Estimation." International Conference on Learning Representations. 2020.\
> [7] Fort S, Hu C H, Lakshminarayanan B. Deep Ensembles: A Loss Landscape Perspective[J]. 2019.\
> [8] Andres R. Masegosa. Learning under model misspecification: Applications to variational and ensemble methods. In Advances in Neural Information Processing Systems, 2020.\
> [9] Zhou, Tianyi, Shengjie Wang, and Jeff A. Bilmes. "Diverse ensemble evolution: Curriculum data-model marriage." Proceedings of the 32nd International Conference on Neural Information Processing Systems. 2018.

---

### Official Review · Reviewer_vevN · 2021-11-04

**Correctness:** 4
**Technical Novelty And Significance:** 2
**Empirical Novelty And Significance:** 3
**Recommendation:** 5
**Confidence:** 3

**Main Review:**

I don't work in this area, so I may be not familiar with the related work. From my viewpoint, using the performance of base learner to train a specialized weak learner is a quite natural idea. For the method part, I have one question:

1. It is not quite reasonable to use a binary cross entropy to supervise w in Eqn 3. Think that one easy example can be correctly classified by all the base classifiers, then all the base learner will try to fit it. This is contradict to the motivation of specialization.

For the experiments, I have the following two questions:
1. Comparisons with Boosting tree related methods are unfair. The base learner is quite different. According to Table1, even the single model is better than the boosting tree methods, which makes this comparison meaningless. If for a fair comparison, the authors should provide comparisons of boosting and various ensemble based on the same base learner.
2. For the image experiments, The baseline and dataset is too weak to demonstrate the superiority. I suggest using the most recognized combination ResNet50 +ImageNet for this experiment.



**Summary Of The Paper:**

This paper presents a novel method for ensemble learning. By introducing a anchor scheme and specialization loss , the base learner are forced to be specialized. The method is validated on both tabular and image datasets.

**Summary Of The Review:**

To summarize, I think this paper does not contain many new concept from my view. According to the experiments, I think the results are good, however some experiments are not that solid. Since I don't work in this area, I rate this paper as borderline reject, and would like to participate the discussion in rebuttal.

---

> ### Author Response · Authors · 2021-11-19
> **Response to Reviewer vevN**
>
> Thanks for your constructive suggestions and comments, which are very helpful for us to improve this paper. Next, we would like to respond to each of the concerns raised in your comments.
>
> > **Q1. "From my viewpoint, using the performance of base learner to train a specialized weak learner is a quite natural idea."**
>
> While it is a natural idea, we are the first work to explicitly enforce sample-wise and model-wise specialization, and obtain promising performance.
>
> Perceiving different models' performance is not a trivial work, hindering ensemble from achieving either sample-wise or model-wise specialization. Previous approaches, such as [1], assign base models with different tasks (e.g., different sample subsets) for training *based on their performance*, however, prediction aggregation is done by simple voting or averaging without considering base models' capabilities.
>
> Instead, SANE fulfills this idea by combining specialization-enhanced training of the base model with a specialization-aware ensemble, and contributes to solving critical challenges of the existing ensemble strategies, which has also been mentioned in the comments of the reviewer y2rX. We believe this new framework can help to inspire other researchers to design ensemble learning methods from a new (and maybe more promising) perspective.
>
> > **Q2. "It is not quite reasonable to use a binary cross entropy to supervise w in Eqn 3. Think that one easy example can be correctly classified by all the base classifiers, then all the base learner will try to fit it. This is contradict to the motivation of specialization."**
>
> ***The choice of cross entropy:*** Assigning samples to $K$ models based on their specialization is equal to solving a $K$-class classification task with soft supervision. Therefore, in Eq. (3), we supervise sample-specific model weights $(\mathbf{w})$ using the cross-entropy loss commonly adopted for classification tasks.
>
> ***Easy sample case:*** In this case, the ensemble model can produce correct predictions by aggregating the predictions from base learners similar to simple average. Since easy samples can be correctly predicted by all base learners, using binary cross-entropy loss to supervise is almost equivalent to uniformly sampling their predictions, because all the base learners could be equally confident to predict them.
>
> The existence of such samples indicates that the specialties of the base models may have overlaps (e.g., due to too many base models or poor specialization) but does not necessarily indicate the failure of specialization. Actually, as demonstrated in Figure 3 in the paper, most of the samples are indeed specialized by one base model only, indicating that SANE can learn specialized base models using the proposed learning objective.
>
> > **Q3. "Comparisons with Boosting tree related methods are unfair."**
>
> Boosting tree is widely known as a promising method on tabular datasets. We follow the baseline settings used by [2] to conduct the comparisons. The taken results of XGBoost and LightGBM for Higgs Boson have been evaluated on the same test set as ours, with exhaustive hyper-parameter optimization [3], and we believe it is promising in our comparison of this paper.
>
> > **Q4. "I suggest using the most recognized combination ResNet50 +ImageNet for this experiment."**
>
> Thank you for the constructive suggestion. Since we are more focused on evaluating the effectiveness and generalization of the framework on different data types, and in order to compare with previous works mainly focusing on CIFAR datasets, some experiments may have been missed. Following your advice, we have added experiments on Tiny ImageNet. We carefully considered your suggestion of using ResNet-50 as the backbone. However, since [4] showed that EfficientNet-B0 outperforms ResNet-50 on ImageNet, we use EfficientNet-B0 (as an alternative) and ResNet-110 in the additional experiments. The results are in Table 1 in the "Reply to All" section and also in Table 4 of the revised manuscript.
>
> References:\
> [1] Zhou, Tianyi, Shengjie Wang, and Jeff A. Bilmes. "Diverse ensemble evolution: Curriculum data-model marriage." Proceedings of the 32nd International Conference on Neural Information Processing Systems. 2018.\
> [2] Zhang, Shaofeng, Meng Liu, and Junchi Yan. "The diversified ensemble neural network." Advances in Neural Information Processing Systems 33 (2020).\
> [3] Anghel, Andreea, et al. "Benchmarking and optimization of gradient boosting decision tree algorithms." arXiv preprint arXiv:1809.04559 (2018).\
> [4] Tan, Mingxing, and Quoc Le. "Efficientnet: Rethinking model scaling for convolutional neural networks." International Conference on Machine Learning. PMLR, 2019.

---

> ### Comment · Reviewer_vevN · 2021-11-30
> **Response**
>
> For Q2, softmax or similar loss which introduces competition between base learners makes more sense to me.
> For Q3, I insist the comparison is not fair since the base learners are different. These results can only illustrate the usefulness of single base learner chosen, not the ensemble method itself.
>
> Above all, I will keep my rating here.

---

> > ### Author Response · Authors · 2021-11-30
> > **Response to Reviewer verN**
> >
> > Thank you for your reply. We want to address your concerns as follows.
> >
> >
> >
> > > **Q2.**
> >
> >
> > First, we would like to emphasize that under soft supervision, the ground truth of the $K$-class classification is the predicted probability of different models for the true labels, rather than the one-hot labels indicating their correctness. Thus, by computing the cross-entropy loss between the softmaxed values of the predicted model weights and the true confidences, we encourage them to specialize in what they can do well.
> >
> >
> > In other words, the direction advocated by SANE is specialization rather than competition. For the ensemble of different model predictions, in Eq. (4), we apply softmax to the predicted model weights so that we can enhance the performance by encouraging the correct models and discouraging the incorrect ones to "participate".
> >
> >
> > > **Q3.**
> >
> >
> > By comparing with the random ensemble, we can see that SANE works better with **same base learner**, proving the usefulness of our ensemble approach. In addition, we compared with TabNet, which was shown to perform better than tree-based methods on tabular datasets. With reference to previous papers on ensemble learning [2], the comparison are fair for illustrating the effectiveness of our ensemble method. Tree-based methods are used as references, while we focus on the neural-network-based approach and showcase our effectiveness on ensemble learning. Our extensive experiments on image datasets can also prove the effectiveness.

---

> > > ### Comment · Reviewer_vevN · 2021-11-30
> > > **response**
> > >
> > > For Q3, the experiments can only validate your method is better than random ensemble, not advanced methods like boosting, etc.

---

> > > > ### Author Response · Authors · 2021-11-30
> > > > **Response to Reviewer vevN**
> > > >
> > > > In this paper, we focus on neural-network-based ensemble learning. The boosting-based ensemble is hard to parallelize and can lead to long training time when applied to deep neural networks [1].
> > > >
> > > > We thank you for your comments for the discussion about the experiments on tabular data. We will refine the description for alleviating potential misunderstanding in the latest version.
> > > >
> > > >
> > > >
> > > > References:\
> > > > [1] Zhou, Tianyi, Shengjie Wang, and Jeff A. Bilmes. "Diverse ensemble evolution: Curriculum data-model marriage." Proceedings of the 32nd International Conference on Neural Information Processing Systems. 2018.

---

### Author Response · Authors · 2021-11-19
**Reply to all -- Experimental results on Tiny ImageNet**

We appreciate all the reviewers' insightful comments. Since most reviewers raised concerns about the effectiveness of SANE on "more complex" (Reviewer y2rX) datasets with "more recognized" (Reviewer vevN) and 'big' (Reviewer G9Xj) models, we conducted additional experiments on Tiny ImageNet to address this common concern.

While most of the previous papers focused on the CIFAR dataset, Snapshot Ensemble [1] conducted experiments on Tiny ImageNet using ResNet-110 as the backbone network, which we followed. In addition, experiments on EfficientNet (EfficientNet-B0 and -B1), a promising choice on the dataset [2], were performed following the setting of [3]. We compared with Random Ensemble, Snapshot Ensemble, and Fast Ensemble [4], and the results are as follows.

| Backbone          | EfficientNet-B0 | EfficientNet-B1 |ResNet-110 |
|-------------------|-----------------|--------------|--------------|
| Single model      | 59.79 $_{\pm \textit{ }0.36}$          | 58.54 $_{\pm \textit{ }0.27}$       |54.67 $_{\pm \textit{ }0.30}$       |
| Fast ensemble     | 60.56 $_{\pm \textit{ }0.15}$            | 61.85 $_{\pm \textit{ }0.14}$       |60.66 $_{\pm \textit{ }0.86}$       |
| Snapshot ensemble | 64.28 $_{\pm \textit{ }0.24}$          | 64.85 $_{\pm \textit{ }0.16}$       |60.32 $_{\pm \textit{ }0.69}$       |
| Random ensemble   | 64.53 $_{\pm \textit{ }0.13}$           | 64.59 $_{\pm \textit{ }0.45}$       |61.45 $_{\pm \textit{ }0.03}$       |
| SANE              | **65.27** $_{\pm \textit{ }0.07}$           | **65.31** $_{\pm \textit{ }0.06}$        |**61.91** $_{\pm \textit{ }0.24}$       |

From the results, SANE works on larger datasets as well and consistently outperforms baselines. These experimental results have been updated in Appendix B of the revised manuscript. In addition, experiments on the entire ImageNet, which is very time-consuming, will be included in the final version of the paper.

References:\
[1] Huang G, Li Y, Pleiss G, et al. Snapshot ensembles: Train 1, get m for free[J]. arXiv preprint arXiv:1704.00109, 2017.\
[2] https://paperswithcode.com/sota/image-classification-on-tiny-imagenet-1 \
[3] Luo Y, Wong Y, Kankanhalli M, et al. Direction concentration learning: Enhancing congruency in machine learning[J]. IEEE transactions on pattern analysis and machine intelligence, 2019.\
[4] Garipov, Timur, et al. "Loss surfaces, mode connectivity, and fast ensembling of dnns." Proceedings of the 32nd International Conference on Neural Information Processing Systems. 2018.

---

### Decision · Program_Chairs · 2022-01-20

**Decision:**

Reject

**Comment:**

This paper proposes a novel ensemble method that enforces specification of the base models to improve accuracy. Base models are specialized on sub-regions of the latent space. To calculate the ensemble prediction for a given example base models are weighted based on how close the example embeddings are to a learned “anchor” embedding. To derive the correlation between samples and anchors,  transformer-like attention mechanism is used.

Reviewers pointed out limitations in the experimental analysis. In turn authors added experiments on tiny-image net with additional architectures and a comparison to several additional baselines on CIFAR-10/100 supporting their findings, which improved the paper. Nevertheless, the paper remained on the borderline after the discussion period and reviewers continued to have doubts about the the significance and novelty of the proposed method, therefore the paper can not be accepted in its current form.